# Impact of frailty in elderly patients with moderate to severe asthma

**Ricardo G. Figueiredo** [1,2,3] *, **Gabriela P. Pinheiro**[2], **Vanessa Arata**[1], **Maisa F. M. Leal**[2], **Cinthia V. N. Santana**[2], **Taciana L. Tiraboschi**[1,3], **José Bessa Junior**[1,3], **Álvaro A. Cruz**[2,4]

**1** Departamento de Saúde, Universidade Estadual de Feira de Santana, Feira de Santana, Bahia, Brazil, **2** Fundação ProAR, Salvador, Bahia, Brazil, **3** Programa de Pós-Graduação em Saúde Coletiva, Universidade Estadual de Feira de Santana, Feira de Santana, Bahia, Brazil, **4** Faculdade de Medicina, Universidade Federal da Bahia, Salvador, Bahia, Brazil

☯ These authors contributed equally to this work.

* rgfigueiredo@uefs.br

## Abstract

Frailty assessment has been identified as critical approach in chronic respiratory diseases with substantial impact in the health status and functionality in later life. Aging modifies the immune response leading to a chronic pro-inflammatory state and increased susceptibility to airway infections. Since epigenetic changes, airway epithelium dysfunction and inflammatory cytokine activity seem to be more pronounced in the immunosenescence, elderly asthmatics are at higher risk of poor clinical outcomes. Therefore, we hypothesize that frailty would be associated with the degree of asthma control in elderly patients with moderate to severe asthma. The aims of this study are to investigate association between frailty and asthma control in patients over 60 years old to estimate the prevalence of frailty in this study population. We plan to conduct a cross-sectional study with at least 120 patients above 60 years old with diagnostic of moderate to severe asthma according to Global Initiative for Asthma (GINA) guidelines, treated at a referral outpatient clinic. We defined asthma control by the six-domain Asthma Control Questionnaire (ACQ-6) and frailty phenotype in accordance with Fried scale and visual scale of frailty (VS-Frailty). We hope to analyze the multidimensional relationships between frailty and asthma and contribute to innovative therapeutic plans in geriatric asthma.

## Introduction

Aging is an ongoing process that modifies the immune response leading to a chronic pro-inflammatory state and increased susceptibility to airway infections, since epigenetic changes, airway epithelium dysfunction and inflammatory cytokine activity seem to be more pronounced in the immunosenescence [1]. Although gene expression patterns might be related to environmental exposure, nutrition and lifestyle over time, senescence is a natural biological process in which an individual experiences the decline of physiologic reserve and it is intrinsically related to aging. The threshold of health maintenance and development of illnesses can

**Data Availability Statement:** No datasets were generated or analysed during the current study.

**Funding:** Article processing charges was funded by Programa de Pós-graduação em Saúde Coletiva da Universidade Estadual de Feira de Santana

(PPGSC-UEFS) The authors received no other specific funding for this work.

**Competing interests:** The authors have declared that no competing interests exist.

be crossed more rapidly in frail patients with a decline in the ability to overcome metabolic stress of any cause, increasing morbidity and mortality [2].

The concept of frailty was initially related to functional decline and comorbidities, regardless of aging [3]. Currently, frailty encompasses a more plural concept, recognized as a multidimensional syndrome characterized by cumulative multisystemic functional decline [4]. This is a multifactorial process with a complex pathophysiology and imbricated organic interrelationships, evolving physical, psychosocial, and economic determinants. Frailty is related to instability and risk of functional loss due to a low resilience of the system to react to biological stress [5, 6], and increased mortality [7]. According to the Cardiovascular Health Study, the prevalence of frailty among elderly was approximately 7%, and the criteria for the frailty phenotype were outlined as shrinking, weakness, poor endurance and energy, slowness and/or low physical activity level [3]. Frailty generally becomes more evident in advanced stages and early recognition of this syndrome may be challenging.

Frailty definition can be often associated with chronic illnesses, and it becomes essential to individualize this syndrome as a completely separated one whose clinical outcomes are of extreme relevance [8, 9]. Indeed, frailty and chronic illnesses have substantial impact in the health status and functionality in later life [10]. Frailty was associated with a 3-fold increased risk of death compared with robust older adults over a period of three years [11]. Other outcomes such as hospitalization and falls were significantly higher in frail than in non-frail patients.

Frail elderly may experience a range of physiologic disfunctions in swallowing ability, increasing the risk of aspiration, and choking [12]. Additionally, functional oropharyngeal dysphagia is highly prevalent among vulnerable older people and may lead to exacerbations in patients with chronic respiratory diseases, such as asthma and chronic obstructive pulmonary disease (COPD) [13]. Sarcopenia affects posture and chewing, therefore interfering with maintenance of vertical position of the head and causing exhaustion in frail elderly in contrast to robust individuals [14]. Given the weakness in chewing, the tendency is to offer softer texture food which often are poorer in nutritional density, favoring to a chain of negative feedback on weight and strength.

Accumulating evidence indicates that frailty may be a critical prognostic factor in patients with chronic respiratory diseases [15]. Notably, frailty status was more intimately associated to mortality than decline in lung function. Frail COPD patients had higher hospitalization rates and falls, and decreased ability to engage in the pulmonary rehabilitation [16]. Concomitant chronic inflammation and changes in the ventilatory mechanical can deteriorate respiratory muscular function, especially in clinically advanced disease [17]. As a result, patients are more likely to present sarcopenia, swallowing disorders and deteriorated functional status [18]. Frail individuals have also failed to show improvement of exercise capacity in pulmonary function, and they expend more energy in the same activity compared to non-frail ones [15].

Asthma in the elderly is an emerging public health issue and age, depression and smoking are associated with decreased survival [19]. Older patients may experience a poor perception of asthma symptoms and partially non-reversible functional obstruction due airway remodelling [20]. Although older age has been associated with worse lung function, it seems to not be a good predictor of poor control or hospital admissions in a properly treated population [21], and a low degree of patient-physician concordance regarding the importance of the symptoms reported has been described in elderly asthmatics [22]. Additionally, accurate interpretation of lung function tests can be challenging because physiologic age-related respiratory changes mimic obstructive functional patterns and neurocognitive function could eventually impair the ability to perform spirometry in some patients [23]. Furthermore, epigenetic mechanisms mediate nonstructural changes in gene expression that are regulated by DNA methylation

patterns, microRNA expression and histone modifications have been described with advancing age [24]. It is possible that some of the changes noted above might be associated with airway remodelling and poor treatment response. Several studies have found higher frequencies of depression in asthmatics, especially in uncontrolled disease [25].

Notably, exercise capacity depends on an integrated response of peripherical muscles and cardiorespiratory endurance. Age-associated changes, physical inactivity, and comorbidities may impair several physiological processes that lead to exercise limitation in the senescence, such as pulmonary mechanics and respiratory muscle function, muscle metabolism, gas transport and diffusion, regulation of muscle blood flow and ventilatory response during exercise, pulmonary vascular function, and cardiac output [26].

The complex interactions of asthma control in frail patients have not been fully understood and very little original research in this field was published in the last few years. A Japanese study including 69 older adults with asthma reported a frailty prevalence of 14,5% and a significant association between frailty and patient-reported [27]. Several comorbidities associated with asthma in the elderly usually diverge from those observed in younger patients, such as cardiac disfunction, obstructive sleep apnea and osteoporosis [23]. Although asthma is a highly prevalent chronic disease, to our knowledge, the prevalence of frailty among elderly patients with moderate-severe asthma is still unknown. Therefore, a comprehensive approach to the underlining comorbidities and multidimensional aspects of frailty are key factors for proper management of asthma in advanced age.

## Hypothesis

In elderly patients with moderate to severe asthma treated in a tertiary center in the city of Salvador, frailty will be associated with the degree of asthma control.

## Objectives

### Primary objective

To investigate association between frailty and asthma control in patients over 60 years old.

### Secondary objectives

1. To estimate the prevalence of frailty in patients over 60 years old with moderate to severe asthma and to report a measure of association between asthma control and frailty

2. To compare frailty prevalence using Fried frailty scale and visual scale of frailty (VS-Frailty)

## Materials and methods

We will conduct a cross-sectional study with at least 120 patients above 60 years old with diagnostic of moderate to severe asthma according to Global Initiative for Asthma (GINA) guidelines, treated at the central referral outpatient clinic of Programa de Controle da Asma e Rinite Alérgica na Bahia (ProAR, Bahia State Asthma and Allergic Rhinitis Control Program) in the city of Salvador, Bahia state, Brazil. The diagnosis of asthma and the classification of its severity will be in accordance with the criteria established by GINA 2022 [28].

Asthma control (dependent variable) will be defined by the six-domain Asthma Control Questionnaire (ACQ-6), validated in the Portuguese language for use in Brazil, based on reported asthma symptoms and the use of rescue B2-agonist medication in the last seven days. Scores for the ICQ-6 range from 0 to 6, with higher scores reflecting poor asthma control. An

ACQ-6 score with cut-off $\geq$ 1.5 defines uncontrolled asthma, while a score $\leq$ 0.75 means adequate asthma control [29].

The diagnosis of frailty (independent variable) will be defined in accordance with Fried frailty scale and visual scale of frailty (VS-Frailty), both validated in the Portuguese language for use in Brazil [11, 30]. Fried scale range from 0 to 5 defined by the analysis of five domains including shrinking, weakness, poor endurance and energy, slowness and/or low physical activity level, where shrinking is at least 5% of weight loss or loss of 10 lbs. in the year before the interview; weakness is hand grip strength in the lowest quintile for age and sex; poor energy is assessed by a self-reported exhaustion questionnaire; slowness is measured by gate speed and low physical activity level is attributed to those who are in the lowest quintile of kilocalories spent according to a score, in the original study the Minnesota Leisure Time (MLTA) was used. Applying the Frail scale, the combination of 3 or more of frailty domains defines the frail phenotype; 1 or 2 criteria are categorized as prefrail, and the absence of any of the criteria is the non-frail group or robust [2].

Hand strength will be evaluated by measuring hand grip strength, in kg/force, using a BASELINE calibrated hydraulic dynamometer, model ER Hi-Res. Three measurements will be collected by a trained team, with a minimum interval of 1 minute, in the dominant hand, in a sitting position with 90º of elbow flexion [31]. The best of the three consecutive measurements will be considered for analysis.

The study was approved by the ethics institutional review board of Universidade Estadual de Feira de Santana (CAAE: 3.505.830–07/29/2019) following the ethical principles of the Declaration of Helsinki. Written informed consent will be obtained from patients before inclusion in the study. We intend to make research data freely available upon request of other investigators and study participants.

## Study population

In a non-probabilistic sample, all patients will be approached by the research staff and invited to participate in the study. Those who meet the eligibility criteria of age over 60 years old, diagnosis of asthma in accordance with the criteria established by GINA and agree to participate in the study will be consecutively enrolled.

Based on a recent comparative analysis between patients with asthma conducted at ProAr (Salvador-Bahia) and Europeans from the U-BIOPRED database [32], we expect a greater proportion of a race or ethnic group other than white and females in our study sample

## Exclusion criteria

1. Diagnosis of other pulmonary morbidities or extrapulmonary disease that could possibly interfere in the evaluation of asthma

2. Asthma exacerbation in the last 4 weeks

3. > 10 pack-year smoking

## Variables, confounders, and effect modifiers

Demographic characteristics, occupational status, smoking behavior, and comorbidities will be obtained during the interview. Clinical data regarding body mass index (BMI), asthma treatment, medication compliance, use of oral corticosteroids, inhaler technique, grip strength, history of exacerbations and hospitalizations will also be evaluated. Spirometry and flow-volume curves, before and after bronchodilators, will be performed according to the specifications

of the American Thoracic Society (ATS) in a computerized spirometer (Koko Spirometer, PDS Instrumentation, Louisville, USA). For this purpose, the best among three reproducible values, in acceptable curves, and with an amplitude of less than 5%, will be recorded as baseline forced expiratory volume in the first second (FEV1).

We plan to analyze the effect of age, gender, dementia, depression, and history of exacerbation as potential confounders. We also plan to investigate the impact of BMI as an effect modifier.

## Quality control

ACQ-6 questionnaire has been validated for Brazilian Portuguese; it will be measured by trained researcher blinded to frail scales. Fried scale has also been validated for Brazilian Portuguese; it will be measured by trained researcher blinded to asthma control questionnaire. Spirometry will be performed by trained staff.

## Sample size

We plan originally for this study to include 120 patients to evaluate our primary outcome. We calculated that this study population would have 80% power to detect a true increase of 20 percent between both groups. A recent Japanese study using the same methodology with 69 patients with asthma over 65 years old have found a frailty prevalence of 15.4% [26].

## Data analysis

We intend to use IBM SPSS statistics to perform statistical analysis. Descriptive statistics will be used to summarize the demographic characteristics of study participants. Categorical variables (Frailty scales) will be expressed as numbers and proportions. Continuous variables (ACQ-6) will be expressed as mean, standard deviation, median and quartiles, and compared with the t-test, t-test for paired samples, Mann-Whitney or Wilcoxon U test. The Chi-square test will be used to compare categorical variables. Kruskal-Wallis test will be used to compare categorical independent variable (frailty status) and continuous dependent variable (ACQ-6). We plan to use logistic regression to estimate the OR (95% CI) for frailty and control and analyze the impact of potential confounders. We plan to use a Roc curve to compare CHS and visual analog scales. We assume a $\alpha$ error of less than 0.05 as statistically significant. A beta error of less than 20% will be assumed for sample calculation used (Power$>$ = 80%).

## Strengths and limitations

To our knowledge, this will be the largest study access the prevalence of frailty and the relationships between frailty and disease control in moderate-severe asthma patients. There is a higher prevalence of chronic diseases in elderly. Comorbidities as dementia and depression are potential confounders that may affect asthma control. Moreover, Asthma and COPD (COPD) are prevalent conditions in older age, both characterized by the presence of airflow obstruction, and might even overlap in some patients [33].

## Risks

The data obtained from the medical record and from the medical interview will be kept confidential and restricted. Access to the Redcap database will be limited to principal investigators. Although best effort will be mobilized to ensure anonymity of all records, there is a potential risk of accidental breach of anonymity. During COVID-19 pandemic, patient and staff safety will be a major concern. Research biosafety requirements will be guarantee including the staff

training, personal protect equipment use and social distancing. The executive research committee decided to avoid lung function tests in patients without clinical indication.

## Impact

According to the World Health Organization (WHO), there were 703 million persons aged 65 years or over in the world in 2019. The number of older persons is projected to double to 1.5 billion in 2050. Indeed, the prevalence of asthma in this population range from 7.0 to 10.6% with perturbing high asthma-related mortality [34]. Furthermore, the multidimensional relationships between frailty and asthma have a major impact on clinical management with positive implications for innovative therapeutic plans for the elderly. Evidence related to asthma control in frail patients may also clarify the complex biological pathways that modulate the susceptibility to exacerbations and reduced quality of life in this population.

## Author Contributions

**Conceptualization:** Ricardo G. Figueiredo, Vanessa Arata, Taciana L. Tiraboschi, José Bessa Junior, Álvaro A. Cruz.

**Data curation:** Ricardo G. Figueiredo, Vanessa Arata, Maisa F. M. Leal, José Bessa Junior, Álvaro A. Cruz.

**Formal analysis:** Ricardo G. Figueiredo, José Bessa Junior, Álvaro A. Cruz.

**Funding acquisition:** José Bessa Junior, Álvaro A. Cruz.

**Investigation:** Ricardo G. Figueiredo, Vanessa Arata, Maisa F. M. Leal, Cinthia V. N. Santana, José Bessa Junior, Álvaro A. Cruz.

**Methodology:** Ricardo G. Figueiredo, Taciana L. Tiraboschi, José Bessa Junior, Álvaro A. Cruz.

**Project administration:** Ricardo G. Figueiredo, Gabriela P. Pinheiro, Maisa F. M. Leal, Cinthia V. N. Santana, José Bessa Junior, Álvaro A. Cruz.

**Resources:** Ricardo G. Figueiredo, Gabriela P. Pinheiro, José Bessa Junior, Álvaro A. Cruz.

**Software:** José Bessa Junior, Álvaro A. Cruz.

**Supervision:** Ricardo G. Figueiredo, Cinthia V. N. Santana, José Bessa Junior, Álvaro A. Cruz.

**Validation:** José Bessa Junior, Álvaro A. Cruz.

**Visualization:** José Bessa Junior, Álvaro A. Cruz.

**Writing – original draft:** Ricardo G. Figueiredo, Vanessa Arata, José Bessa Junior, Álvaro A. Cruz.

**Writing – review & editing:** Ricardo G. Figueiredo, Gabriela P. Pinheiro, Vanessa Arata, Maisa F. M. Leal, Cinthia V. N. Santana, Taciana L. Tiraboschi, José Bessa Junior, Álvaro A. Cruz.

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
