## [Decision Letter · Decision Letter 0]

10 May 2022

PONE-D-22-05511Impact of Frailty in Elderly Patients with Moderate to Severe AsthmaPLOS ONE

Dear Dr. Figueiredo,

Thank you for submitting your manuscript to PLOS ONE. After careful consideration, we feel that it has merit but does not fully meet PLOS ONE’s publication criteria as it currently stands. Therefore, we invite you to submit a revised version of the manuscript that addresses the points raised during the review process.

This manuscript type is "study protocol" and the protocol does not report results or discussion. Dear authors please pay attention particularly to the comments raised by Reviewer 2 and 3 and respond adequately to clarify their doubts.

We look forward to receiving your revised manuscript.

Kind regards,

Claudio Andaloro

Academic Editor

PLOS ONE

Journal Requirements:

Reviewers' comments:

Reviewer's Responses to Questions

**Comments to the Author**

1. Does the manuscript provide a valid rationale for the proposed study, with clearly identified and justified research questions?

Reviewer #1: No

Reviewer #2: Yes

Reviewer #3: Partly

2. Is the protocol technically sound and planned in a manner that will lead to a meaningful outcome and allow testing the stated hypotheses?

Reviewer #1: Yes

Reviewer #2: Yes

Reviewer #3: Yes

3. Is the methodology feasible and described in sufficient detail to allow the work to be replicable?

Reviewer #1: Yes

Reviewer #2: Yes

Reviewer #3: Yes

4. Have the authors described where all data underlying the findings will be made available when the study is complete?

Reviewer #1: No

Reviewer #2: Yes

Reviewer #3: No

5. Is the manuscript presented in an intelligible fashion and written in standard English?

Reviewer #1: Yes

Reviewer #2: Yes

Reviewer #3: Yes

6. Review Comments to the Author

You may also provide optional suggestions and comments to authors that they might find helpful in planning their study.

Reviewer #1: Thank you for the opportunity to evaluate the article. However, this article only includes the study protocol. No information is given about the results and analyzes obtained. Although similar studies are available in the literature, subgroup analyzes according to the severity of asthma may contribute to the literature. However, I still do not think that the protocol alone is sufficient for admission to the journal.

Best regards.

For example: Kusunose M, Sanda R, Mori M, Narita A, Nishimura K. Are frailty and patient-reported outcomes independent in subjects with asthma? A cross-sectional observational study. Clin Respir J. 2021;15(2):216-224. doi:10.1111/crj.13287

Reviewer #2: 1. A brief definition of frailty should be given in the introduction section.

2. The second paragraph in the introduction about frailty should be embedded into materials and methods section.

3. Why was age 60 years chosen as the inclusion criteria? Is there a specific explanation?

4. The statement ‘Asthma in the elderly is associated with disease severity, non-eosinophilic phenotype, and reduced lung function’ should be reviewed. A more well-known manuscript could be cited as reference 10.

5. What were the mentioned comorbidities in older asthmatics that diverge from younger patients?

6. Please be consistent with the year of GINA guideline and referencing.

7. Instead of bullet points please try to prefer plain text if possible.

8. Please comment on disregarded of respiratory compromise due to lowered physical activity in elderly.

9. Reference style (format) should be checked.

10. Please add more keywords.

Reviewer #3: Abstract

The abstract should be written in present tense versus using words like “will”.

Introduction

General comment – greater number of references to back up statements should be provided.

As examples (though there are more):

“Frail elderly may experience a range of physiologic disfunctions in swallowing ability, increasing the risk of aspiration, and choking”.

“As a result, patients are more likely to present sarcopenia, swallowing disorders and

deteriorated functional status.”

Please also expand the number of references for the introduction to enhance rigor of background.

Materials & Methods

Please provide information about the expected diversity of the sample population, including sex and race.

Please provide rationale for the exclusion of “asthma exacerbation in the last 4 weeks”

Please provide a statement about IRB approvals and informed consent. Please also provide information about following principles of Declaration of Helsinki.

Other confounders and something that may be addressed is asthma medication compliance

Other

Although no data are provided the authors should still include a statement should be provided about how the authors plan to share research data from their study when it is completed or published.

7. PLOS authors have the option to publish the peer review history of their article (what does this mean?). If published, this will include your full peer review and any attached files.

Reviewer #1: No

Reviewer #2: No

Reviewer #3: No

---

## [Author Response · Author response to Decision Letter 0]

2 Jun 2022

Dear Dr. Claudio Andaloro, 

Please find attached here the revised version of our manuscript entitled "Impact of Frailty in Elderly Patients with Moderate to Severe Asthma", resubmitted for your consideration for publication in Plos One. You will find our point-by-point responses to the editorial revisions below. All changes in the revised text have been highlighted. 

Please feel free to contact us with any questions or concerns, and we eagerly await your response.

Journal Requirements:

R – We revised the manuscript according to PLOS ONE's style requirements and The PLOS ONE style templates.

R – We have kindly requested the editor office to update your Data Availability statement in a new version of the cover letter, since we will not provide repository information for our data, which we present below:

“We kindly request the editor office to update your Data Availability statement. We will not provide repository information for our data.”

R – Revised accordingly.

“Ricardo G. Figueiredo1,2,3¶*, Gabriela P. Pinheiro2, Vanessa Arata1, Maisa F. M. Leal2, Cinthia V. N. Santana2, Taciana L. Tiraboschi1,3, José Bessa Junior1,3¶, Álvaro A. Cruz2,4¶

1 Departamento de Saúde, Universidade Estadual de Feira de Santana, Feira de Santana, Bahia, Brazil

2 Fundação ProAR, Salvador, Bahia, Brazil

3 Programa de Pós-Graduação em Saúde Coletiva, Universidade Estadual de Feira de Santana, Feira de Santana, Bahia, Brazil

4 Faculdade de Medicina, Universidade Federal da Bahia, Salvador, Bahia, Brazil 

* Corresponding author

E-mail: rgfigueiredo@uefs.br (RGF)

¶These authors contributed equally to this work.

4. Please review your reference list to ensure that it is complete and correct. If you have cited papers that have been retracted, please include the rationale for doing so in the manuscript text or remove these references and replace them with relevant current references. Any changes to the reference list should be mentioned in the rebuttal letter that accompanies your revised manuscript. If you need to cite a retracted article, indicate the article’s retracted status in the References list and also include a citation and full reference for the retraction notice.

R – Reference list and citations were revised accordingly

Reviewer #1: Thank you for the opportunity to evaluate the article. However, this article only includes the study protocol. No information is given about the results and analyzes obtained. Although similar studies are available in the literature, subgroup analyzes according to the severity of asthma may contribute to the literature. However, I still do not think that the protocol alone is sufficient for admission to the journal.

For example: Kusunose M, Sanda R, Mori M, Narita A, Nishimura K. Are frailty and patient-reported outcomes independent in subjects with asthma? A cross-sectional observational study. Clin Respir J. 2021;15(2):216-224. doi:10.1111/crj.13287

R – We thank the reviewer for the careful reading of the manuscript and constructive comments. The multidimensional relationships between frailty and asthma are still not fully understood and may have a major impact on personalized treatment for the elderly. To our knowledge, this study protocol will be the largest study to evaluate the prevalence of frailty and the relationships between frailty and disease control in moderate-severe asthma patients. 

Although we have already included this reference in our manuscript, we have cited Dr. Kusunose study in the introduction which we present below: 

“A Japanese study including 69 older adults with asthma reported a frailty prevalence of 14,5% and a significant association between frailty and patient-reported outcomes.”(line 137)

Reviewer #2

1) A brief definition of frailty should be given in the introduction section.

R - We agree with the reviewer that include frailty definition in the introduction would add relevance for the reader:

“The concept of frailty was initially related to functional decline and comorbidities, regardless of aging [3]. Currently, frailty encompasses a more plural concept, recognized as a multidimensional syndrome characterized by cumulative multisystemic functional decline [4]. This is a multifactorial process with a complex pathophysiology and imbricated organic interrelationships, evolving physical, psychosocial and economic determinants. Frailty is related to instability and risk of functional loss due to a low resilience of the system to react to biostressors [5,6], and increased mortality [7].” (line 74)

2) The second paragraph in the introduction about frailty should be embedded into materials and methods section.

R - Revised accordingly. We allocated the follow paragragh in the methods section and included a new reference for the measure of grip strengh:

“The criteria for the frailty diagnosis were defined as shrinking, weakness, poor endurance and energy, slowness and/or low physical activity level, where shrinking is at least 5% of weight loss or loss of 10 lbs in the year before the interview; weakness is hand grip strength in the lowest quintile for age and sex; poor energy is assessed by a self-reported exhaustion questionnaire; slowness is measured by gate speed and low physical activity level is attributed to those who are in the lowest quintile of kilocalories spent according to a score, in the original study the Minnesota Leisure Time (MLTA) was used. Applying the Frail scale, the combination of 3 or more of frailty domains defines the frail phenotype; 1 or 2 criteria are categorized as prefrail, and the absence of any of the criteria is the non-frail group or robust4. Hand strength will be evaluated by measuring hand grip strength, in kg/force, using a BASELINE calibrated hydraulic dynamometer, model ER Hi-Res. Three measurements will be collected by a trained team, with a minimum interval of 1 minute, in the dominant hand, in a sitting position with 90º of elbow flexion. The best of the three consecutive measurements will be considered for analysis.”

3) Why was age 60 years chosen as the inclusion criteria? Is there a specific explanation?

R - According to The World Health Organization (WHO) report, environments are highly influential on our behaviour and our exposure to health risks (for example, air pollution or violence), our access to services (for example, health and social care) and the opportunities that ageing brings. Indeed, the official definition of elderly may vary among developed and non-develop countries. We will carry out this study in a developing country with a life expectancy of 75 years old. The WHO considered a 60-year-old cut-off in the life expectancy global report. 

Due the intrinsic characteristics of our study population, we decided to use 60 years as the inclusion criteria.

4) The statement ‘Asthma in the elderly is associated with disease severity, non-eosinophilic phenotype, and reduced lung function’ should be reviewed. A more well-known manuscript could be cited as reference 10.

R- We thank the reviewer for the suggestion. We have revised the text which we present below:

“Asthma in the elderly is an emerging public health issue and age, depression and smoking are associated with decreased survival”

Gibson PG, McDonald VM, Marks GB. Asthma in older adults. The Lancet [Internet]. 2010 Sep 4 [cited 2020 May 27];376(9743):803–13.

5) What were the mentioned comorbidities in older asthmatics that diverge from younger patients?

R – A higher prevalence of cardiovascular and metabolic diseases has been described in older adults. We have revised the text which we present below:

“Several comorbidities associated with asthma in the elderly usually diverge from those observed in younger patients, such as cardiac disfunction, obstructive sleep apnea and osteoporosis.”

6) Please be consistent with the year of GINA guideline and referencing.

R - Revised accordingly. Global Initiative for Asthma - Global Initiative for Asthma - GINA [Internet]. Global Initiative for Asthma - GINA. 2022. Available from: http://www.ginasthma.org

7) Instead of bullet points please try to prefer plain text if possible.

R – We added the main caracteristics of Fraity scale and Asthma control questionnaire (ACQ) in the methods section. We also created a new topic entitled “Variables, Confounders and Effect Modifiers”, which we present below:

 “Demographic characteristics, occupational status, smoking behavior and comorbidities will be obtained during the interview. Clinical data regarding body mass index (BMI), asthma treatment, medication compliance, use of oral corticosteroids, inhaler technique, grip strength, history of exacerbations and hospitalizations will also be evaluated. Spirometry and flow-volume curves, before and after bronchodilators, will be performed according to the specifications of the American Thoracic Society (ATS) in a computerized spirometer (Koko Spirometer, PDS Instrumentation, Louisville, USA). For this purpose, the best among three reproducible values, in acceptable curves, and with an amplitude of less than 5%, will be recorded as baseline forced expiratory volume in the first second (FEV1). 

 We plan to analyze the effect of age, gender, dementia, depression, and history of exacerbation as potential confounders. We also plan to investigate the impact of BMI as an effect modifier.”

8) Please comment on disregarded of respiratory compromise due to lowered physical activity in elderly.

R - We allocated the follow paragraph in the introduction section:

“Notably, exercise capacity depends on an integrated response of peripherical muscles and cardiorespiratory endurance. Age-associated changes, physical inactivity, and comorbidities may impair several physiological processes that lead to exercise limitation in the senescence, such as pulmonary mechanics and respiratory muscle function, muscle metabolism, gas transport and diffusion, regulation of muscle blood flow and ventilatory response during exercise, pulmonary vascular function, and cardiac output (ROMAN, 2016). “

9) Reference style (format) should be checked.

R - Reference list and citations were revised accordingly 

10) Please add more keywords.

R - Key words: frailty, asthma, elderly, asthma control, grip strength

Reviewer #3:

We are grateful to the reviewer for their insightful comments on our paper. We have been able to incorporate changes to reflect most of the suggestions provided by the reviewer. Here is a point-by-point response to the reviewers' comments and concerns

1) The abstract should be written in present tense versus using words like “will”.

R - We agree with the reviewer that the abstract should be written in present tense. We have revised the abstract accordingly. 

2) Introduction: General comment – greater number of references to back up statements should be provided.

R - We agree with the reviewer that a greater number of references would add relevance for the reader. We have revised the introduction and added references which we present below:

“Frailty definition can be often associated with disability and chronic illnesses, and it becomes essential to individualize this syndrome as a completely separated one whose clinical outcomes are of extreme relevance (FERRUCCI, 2018).”

“Frail elderly may experience a range of physiologic disfunctions in swallowing ability, increasing the risk of aspiration, and choking (NEY, 2009).”

“Concomitant chronic inflammation and changes in the ventilatory mechanical can deteriorate respiratory muscular function, especially in clinically advanced disease (MACINTYRE, 2016).”

“As a result, patients are more likely to present sarcopenia, swallowing disorders and deteriorated functional status (FUJISHIMA, 2019).”

3) Please provide information about the expected diversity of the sample population, including sex and race.

R – A comparative analysis between patients with asthma conducted at ProAr (Salvador-Bahia) and Europeans from the U-BIOPRED database revealed several ethnic, demographic, socioeconomic and asthma severity differences (CRUZ, 2020). The Brazilian asthmatics showed a greater proportion of a race or ethnic group other than white and females, worse pulmonary function, and poor socioeconomic status.

We have revised the text which we present below:

“Based on a recent comparative analysis between patients with asthma conducted at ProAr (Salvador-Bahia) and Europeans from the U-BIOPRED database, we expect a greater proportion of females and participants of ethnic origin other than in our study sample”

4) Please provide rationale for the exclusion of “asthma exacerbation in the last 4 weeks”

R – Asthma exacerbation may induce a transient period of poor disease control, reduced gait speed and muscle strength. Therefore, ACQ and Fried scale results could be erroneous overestimated right after an exacerbation. 

5) Please provide a statement about IRB approvals and informed consent. Please also provide information about following principles of Declaration of Helsinki.

R – We have added a sentence regarding IRB approvals and informed consent which we present below:

“The study was approved by the ethics institutional review board of Universidade Estadual de Feira de Santana (CAAE: 3.505.830 - 07/29/2019) following the ethical principles of the Declaration of Helsinki. Written informed consent will be obtained from patients before inclusion in the study.”

The IRB approval have been uploaded in PLOS ONE editorial manager

6) Other confounders and something that may be addressed is asthma medication compliance

R - We created a new topic entitled “Variables, Confounders and Effect Modifiers” which we present below:

“Demographic characteristics, occupational status, smoking behavior and comorbidities will be obtained during the interview. Clinical data regarding body mass index (BMI), asthma treatment, medication compliance, use of oral corticosteroids, inhaler technique, grip strength, history of exacerbations and hospitalizations will also be evaluated. (…) We plan to analyze the effect of age, gender, dementia, depression, and history of exacerbation as potential confounders. We also plan to investigate the impact of BMI as an effect modifier.”

7) Although no data are provided the authors should still include a statement should be provided about how the authors plan to share research data from their study when it is completed or published.

R - We included the follow sentence in the methods section:

“We intend to make research data freely available upon request of other investigators and study participants.”

---

## [Decision Letter · Decision Letter 1]

20 Jun 2022

Impact of Frailty in Elderly Patients with Moderate to Severe Asthma

PONE-D-22-05511R1

Dear Dr. Figueiredo,

We’re pleased to inform you that your manuscript has been judged scientifically suitable for publication and will be formally accepted for publication once it meets all outstanding technical requirements.

Kind regards,

Claudio Andaloro

Academic Editor

PLOS ONE

Additional Editor Comments (optional):

Reviewers' comments:

Reviewer's Responses to Questions

**Comments to the Author**

1. Does the manuscript provide a valid rationale for the proposed study, with clearly identified and justified research questions?

Reviewer #2: Yes

Reviewer #3: Yes

2. Is the protocol technically sound and planned in a manner that will lead to a meaningful outcome and allow testing the stated hypotheses?

Reviewer #2: Yes

Reviewer #3: Yes

3. Is the methodology feasible and described in sufficient detail to allow the work to be replicable?

Reviewer #2: Yes

Reviewer #3: Yes

4. Have the authors described where all data underlying the findings will be made available when the study is complete?

Reviewer #2: Yes

Reviewer #3: Yes

5. Is the manuscript presented in an intelligible fashion and written in standard English?

Reviewer #2: Yes

Reviewer #3: Yes

6. Review Comments to the Author

You may also provide optional suggestions and comments to authors that they might find helpful in planning their study.

Reviewer #2: The authors have revised the manuscript and have appropriately addressed the suggestions that I had.

I have no further criticism.

Reviewer #3: The authors have addressed my concerns and the study is now strengthened and improved. I have no additional major concerns.

7. PLOS authors have the option to publish the peer review history of their article (what does this mean?). If published, this will include your full peer review and any attached files.

Reviewer #2: No

Reviewer #3: No

---

## [Editor Report · Acceptance letter]

6 Jul 2022

PONE-D-22-05511R1 

Impact of Frailty in Elderly Patients with Moderate to Severe Asthma 

Dear Dr. Figueiredo:

I'm pleased to inform you that your manuscript has been deemed suitable for publication in PLOS ONE. Congratulations! Your manuscript is now with our production department. 

Kind regards, 

on behalf of

Dr. Claudio Andaloro 

Academic Editor

PLOS ONE